# ACUX Typology: A Harmonisation of Cultural-Visitor Typologies for Multi-Profile Classification

**Markos Konstantakis ***[ID], **Yannis Christodoulou** [ID], **Georgios Alexandridis** [ID], **Alexandros Teneketzis** [ID] **and George Caridakis** [ID]

Department of Cultural Technology and Communication, University of the Aegean, 81100 Mytilene, Greece; yannischris@aegean.gr (Y.C.); gealexandri@aegean.gr (G.A.); ateneketzis@aegean.gr (A.T.); gcari@aegean.gr (G.C.)
* Correspondence: mkonstadakis@aegean.gr

**Abstract:** The modern cultural industry and the related academic sectors have shown increased interest in Cultural User eXperience (CUX) research, since it constitutes a critical factor to examine and apply when presenting cultural content. Recent CUX studies show that visitors tend to carry their own cultural characteristics and preferences when visiting destinations of cultural interest, thus obtaining a virtually unique experience. To cope with this tendency, various research efforts have been made to identify different profiles of cultural visitors based on their background and preferences and classify them into distinct visitor types. In this paper, we proposed the ACUX (Augmented Cultural User eXperience) typology for classifying visitors of cultural destinations. The proposed typology aims to provide the multi-profile classification of cultural visitors based on their visiting preferences. Methodology-wise, the ACUX typology was the output of a harmonisation process of existing cultural-visitor typologies that base their classification on visiting preferences. The proposed typology was evaluated in juxtaposition with the harmonised typologies from which it was derived through an experiment conducted using a recommender and a dataset of TripAdvisor user responses. The evaluation showed that the ACUX typology achieved a more accurate profiling of cultural visitors, enabling them to reduce information overload by directly suggesting content that is more likely to meet their diverse preferences and needs.

**Keywords:** profile clustering; profile alignment; meta-clustering; cultural heritage; personalised recommendation; UNESCO Thesaurus

## 1. Introduction

User eXperience (UX) design is a user interface (UI) design approach for creating digital products that provide meaningful and relevant experiences to users and has grown through time into a vital component of Human-Computer Interaction (HCI). In this context, Cultural User eXperience (CUX) becomes a critical factor when presenting Cultural Heritage (CH) content [1]. For that reason, the cultural industry and relevant academic sectors have shown increased interest in research around CUX [2].

The most recent CUX studies [3–8] show that users of applications and services of cultural content tend to carry their own cultural characteristics and preferences when visiting destinations of cultural interest, learn and interact differently with cultural content, thus obtaining a virtually unique cultural experience. To cope with this tendency, cultural spaces need to search for novel ways of providing more personalised experiences to their visitors [4,9,10]. To that end, various research efforts have been made to identify different profiles of cultural visitors based on their background and preferences, and classify them into distinct visitor types [9,11–16] .

A bibliographic search conducted on existing typologies of cultural visitors shows that the great majority of them assume that destinations of cultural interest correspond to single visitor profiles (rather than multiple) [3–9,11–16]. For example, according to Gibson's typology [8], climbing or ski-jumping correspond to the *Thrill Seeker* profile, who

is interested in extreme sports, but not to the *Explorer* profile, who is *also* bound to enjoy such activities. Furthermore, Sertkan [17] states that existing cultural-visitor typologies are mostly focused on what visitors *believe* they desire, rather than what they would potentially enjoy regardless, while Özel [13] and Chen [15] highlight that existing typologies do not take into account the fact that visitors arrange and experience their trip based on a variety of factors (time availability, income level, health issues, family obligations, technology literacy, social media habits, et al.). Finally, Stylianou [3] claims that the majority of existing cultural-visitor typologies aim to provide visitor profiles that meet the requirements of *specific* cultural spaces, and as such, they are not broadly applicable by default.

Motivated by those findings, in this paper we present the ACUX (Augmented Cultural User eXpierence) typology for classifying visitors of cultural destinations. ACUX provides a *multi-profile* classification of cultural visitors based on their visiting preferences and needs. Methodology-wise, ACUX is the output of a *harmonisation* process of selected cultural-visitor typologies that base their classification on visiting preferences. In particular, the development of the ACUX typology was conducted in the following stages:

- *SELECTION* stage: The selection of cultural-visitor typologies that base visitor classification on visiting preferences (Section 3.2).
- *ALIGNMENT* stage: The alignment of the profiles of the selected typologies according to the *UNESCO Thesaurus* terms for describing cultural resources (Section 3.3).
- *CLUSTERING* stage: The clustering of the profiles of the selected typologies via a three-step clustering process (Section 3.4).

The rest of this paper is structured as follows: Section 2 reviews existing cultural-visitor typologies. Section 3 discusses the ACUX typology development, while Section 4 presents the evaluation of the proposed typology. Finally, conclusions and future research points are drawn in Section 5.

## 2. Related Work

Cultural visitors are often defined in the literature as travellers who visit cultural institutions such as museums, archaeological and historical sites, opera houses, theatres, festivals, etc. [18–25]. Several attempts have been made to classify cultural visitors, as presented below.

Gibson [8] addresses the role of cultural tourist preferences by developing the Tourist Roles Preference Scale (TRPS), a thorough classification of leisure visitors. The latest version of the TRPS examines 15 tourist roles. The author presents statistical evidence of a correlation between tourist roles and life stage, gender, and personal needs. Fifteen tourist roles are distinguished: *Action Seeker, Active Sports Tourist, Anthropologist, Archaeologist, Drifter, Educational Tourist, Escapist, Explorer, High-Class Tourist, Independent Mass Tourist, Jetsetter, Organised Mass Tourist, Seeker, Sun Lover,* or *Thrill Seeker*.

Seaton [11] proposes a number of profiles that may be adopted or emulated by the visitor: *Dilettante/Aesthete, Antiquarian Heritage Seeker, Explorer/Adventurer, Religious Pilgrim and Spiritual Seeker, Festival Charivariist, Littérateur, Epicurean, Natural and Social Scientist, Sportsman, Familiarch, Sun and Sand Hedonist,* or *Modernist*.

Smith [12] suggests a more practical classification of a cultural visitor as: *Heritage Tourist, Art Tourist, Creative Tourist, Urban Cultural Tourist, Rural Cultural Tourist, Indigenous Cultural Tourist,* or *Popular Cultural Tourist*. This typology simplifies the classification of cultural tourists according to the activities they engage in. However, it should be noted that certain groups are not mutually exclusive, for example, as is the case with *Urban, Popular,* and *Creative* cultural tourists. Additionally, the author does not thoroughly address the issue of motivation for cultural tourism or behavioural patterns, as indicated by other researchers [26].

Grün [14] proposes another typology of cultural tourists based on visiting preferences, using an Euclidean metric as a first matchmaking step and a spreading activation algorithm as a second matchmaking step. Grün's profiles are: *Action Seeker, Avid Athlete, Cultural Visitor, Educational Buff, Nature Lover, Sight Seeker,* and *Sun Worshipper*.

Similarly, McKercher [21] developed a typology of cultural visitors, addressing two dimensions: the "centrality (importance) of cultural tourism" (i.e., the degree to which visitors take into account the cultural aspect when choosing a destination) and the depth of cultural experience. In particular, McKercher identifies five types of cultural visitors: *Purposeful, Incidental, Serendipitous, Casual, and Sightseeing*.

McKercher's typology has been extensively employed by scholars studying cultural visitors in different areas. For instance, Kantanen [23] examined the influence of cultural aspects on tourists using McKercher's typology. Furthermore, Vong [20] exploits McKercher's typology to analyse the cultural tourist typologies via a questionnaire survey addressed to incoming tourists in Macau. Finally, Qi [25] refines McKercher's typology and uses it to analyse user-generated content on tourism-related online social media platforms.

Using demographics, travel behaviour, experience, and satisfaction data, De Simone [24] examines the link between tourist typologies and heritage tourist sentiments. Nguyen [19] classified cultural visitors in Vietnam based on two critical variables: the relevance of cultural tourism while visiting a city and the depth of cultural experience. Russell-Rose et al. [27] claim that cultural visitors tend to progress from beginner to expert status gradually through experience. This is defined in further detail in Cifter's [28] "process of gaining experience".

Stylianou [3] reconstructs and builds on well-established cultural visitor typologies while also proposing an alternative model for explaining distinctions between cultural profiles in art museums. The objective of the research was to identify different ways of experiencing a visit to an art museum, whether on-site or virtually. The eight MPFs (Museum Perceptual Filters) identified in the study were: *Professional, Art Loving, Self Exploration, Cultural Tourism, Social Visitation, Romantic, Rejection, and Indifference*. Five case studies were examined, which demonstrated the need for more inclusive and flexible typologies.

Neidhardt [29] proposes a novel way of indirectly eliciting user preferences for cultural visitors via a picture-based approach. The author presents a model in which the visitor's profile is composed of seven critical variables: *Sun-Loving and Connected, Educational, Independent, Culture-Loving, Open-Minded and Sportive, Risk-Seeking, and Nature and Silence-Loving*.

Özel [13] makes a classification of cultural tourist profiles based on motivations, demographic and travel behaviour characteristics. According to Özel, not all cultural tourists have the same degree of interest in the cultural experience of tourism. Thus, a new approach may provide benefits by combining both cultural and non-cultural components while offering cultural tourism products. The profiles derived were: *Relaxation Seeker, Sports Seeker, Family-Oriented, Escapist, and Achievement and Autonomy Seeker*.

Finally, Sertkan [17] proposes an alternative picture-based typology using a factor analysis of pictures with varimax rotation. In particular, the author proposes 17 tourist roles: *Sun and Chill Out, Knowledge and Travel, Independence and History, Culture and Indulgence, Social and Sports, Action and Fun, and Nature and Recreation*.

## 3. ACUX Typology Development

### 3.1. Methodology

The development of the ACUX typology was based on the harmonisation of existing typologies of cultural visitors. The harmonisation process led to the specification of the ACUX profiles (Section 3.5) and was conducted in three stages (see Figure 1):

1.  *SELECTION* stage: The selection of cultural-visitor typologies that base visitor classification on visiting preferences (Section 3.2);
2.  *ALIGNMENT* stage: The alignment of the profiles of the selected typologies according to the *UNESCO Thesaurus* terms for describing cultural resources (Section 3.3);
3.  *CLUSTERING* stage: The clustering of the profiles of the selected typologies via a three-step clustering process (Section 3.4);
    3.1.  The clustering of the profiles aligned in the *ALIGNMENT* stage (Section 3.4.1);
    3.2.  The meta-clustering of the clusters produced in the previous step (Section 3.4.2);

3.3.  The clustering of the profiles that remained unaligned during the *ALIGNMENT* stage (Section 3.4.3).

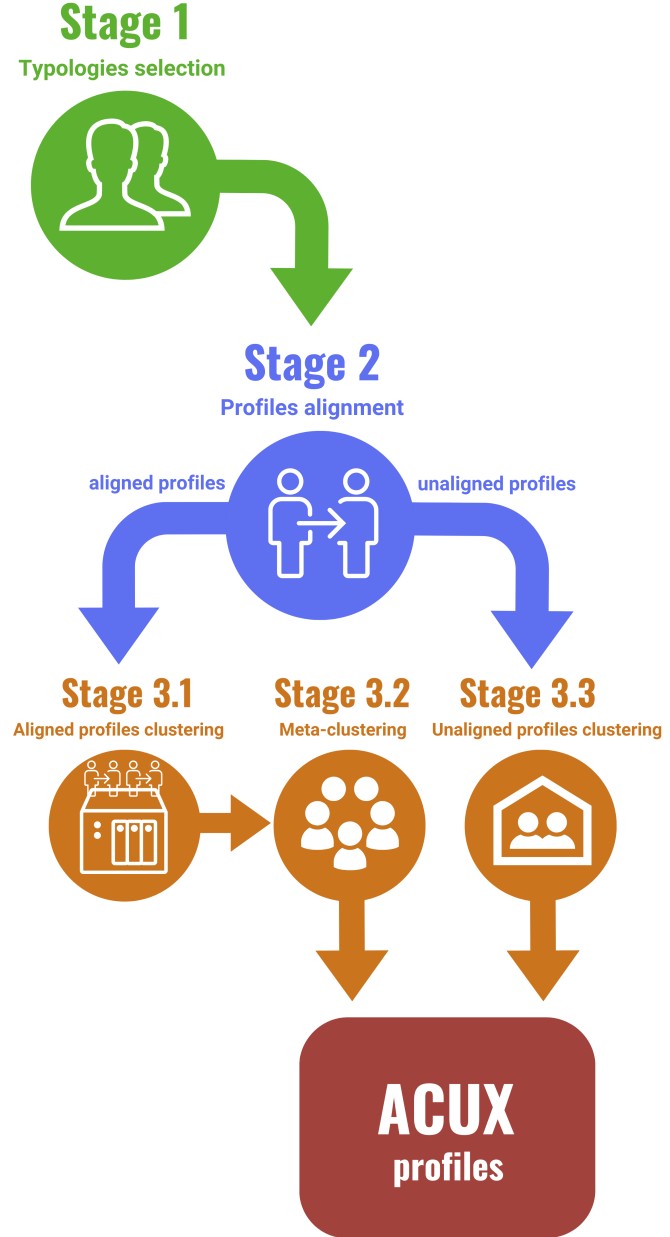

**Figure 1.** ACUX typology development stages.

*3.2. Selection Stage*

As presented in Section 2, numerous research efforts have been made to classify visitors of cultural destinations. Some researchers [13,30] classify cultural visitors according to their motivations, demographics, and travel behaviour characteristics. Others emphasise the importance of cultural tourism and the depth of cultural experience [20,25]. Nevertheless, most researchers claim that the effectiveness and reliability of a typology of cultural visitors should be based on *visiting preferences* as the primary classification criterion [8,11,12,14,21,31–34].

Drawing on the above, in the *SELECTION* stage, we studied existing cultural-visitor typologies by thoroughly examining their profile descriptions in order to distinguish those that base their classification primarily (if not exclusively) on visiting preferences. Next, we only selected those that are considered *primordial*, in the sense that all other

typologies constitute some sort of combination or modification of them. As a result, five typologies were selected to be used in the harmonisation process: Gibson's [8], Seaton's [11], Smith's [12], Grün's [14], and McKercher's [21].

As a last step, we reviewed the profile descriptions of the five selected typologies in order to detect profiles that referred entirely to classification criteria other than those of visiting preferences. In particular, two such cases were identified, and the relevant profiles were excluded from the *harmonisation* process:

(a) Profiles that referred to visitor groups rather than individuals, namely the *Familiarch* profile from Seaton's typology and the *Organised Mass Tourist* and *Independent Mass Tourist* profiles from Gibson's typology;

(b) Profiles that referred to the visitor's economic/social status, namely the *High-Class Tourist* and *Jetsetter* profiles from Gibson's typology.

*3.3. Alignment Stage*

The objective of the *ALIGNMENT* stage was to align the profiles of the five selected typologies. Profile alignment was achieved by using a set of terms defined in the *UNESCO Thesaurus* [35] as a reference point for describing cultural resources. The terms that were utilised for the profile alignment are presented below, together with indicative examples of their narrower concepts:

- *Philosophy and ethics* (e.g., philosophy, metaphysics, moral values, positivism, moral concepts, and rationalism);
- *Literature* (e.g., epic poetry, book reviews, comics, novels, proverbs, speeches, and writers);
- *Entertainment* (e.g., festivals, sporting events, carnivals, zoos, swimming places, motorsport events, indoor games, hill tribes, desert or mountain trekking venues, and climbing and motorsport events);
- *Cultural policy and planning* (e.g., cultural agents, cultural finance, industrial heritage, customs and traditions, folklore, and oral tradition);
- *Leisure* (e.g., entertainment, holidays, indoor games, sports, and swimming);
- *Performing arts* (e.g., cinema, dance, films, operas, popular music, theatre, and vocal music);
- *History* (e.g., antiquity, archaeology, oral history, mythology, and genealogy);
- *Museums* (e.g., historical museums, ecomuseums, art galleries, national museums, archaeological museums, and cultural exhibitions);
- *Religion* (e.g., religious buildings, churches, synagogues, temples, witchcraft, and zoroastrianism);
- *Art* (e.g., architects, artists, folk art, modern art, sculptors, painters, contemporary art, film makers, iconography, and religious art);
- *Visual arts* (e.g., graphic arts, handicrafts, ceramic art, jewellery, engraving, mosaics, pottery, and textile arts);
- *Linguistics* (e.g., alphabets, acronyms, Braille, dialects, foreign languages, semantics, spelling, symbols, terminology, and translation);
- *Culture* (e.g., cultural environment, cultural history, mass culture, cultural minorities, cultural identity, Mayas, and multiculturalism);

The profile alignment of the five selected typologies was carried out by thoroughly examining the description of each profile in juxtaposition with the *UNESCO Thesaurus* terms, also taking into account their narrower concepts. The result of the alignment process is depicted in Figure 2 and summarised in Figure 3. As a final step, profiles that could not be matched with any UNESCO term were detected and marked for subsequent processing (see Section 3.4.3). Specifically, McKercher's *Sightseeing*, Seaton's *Epicurean*, and Grün's *SightSeeker* remained unaligned (highlighted with a red frame in Figure 2).

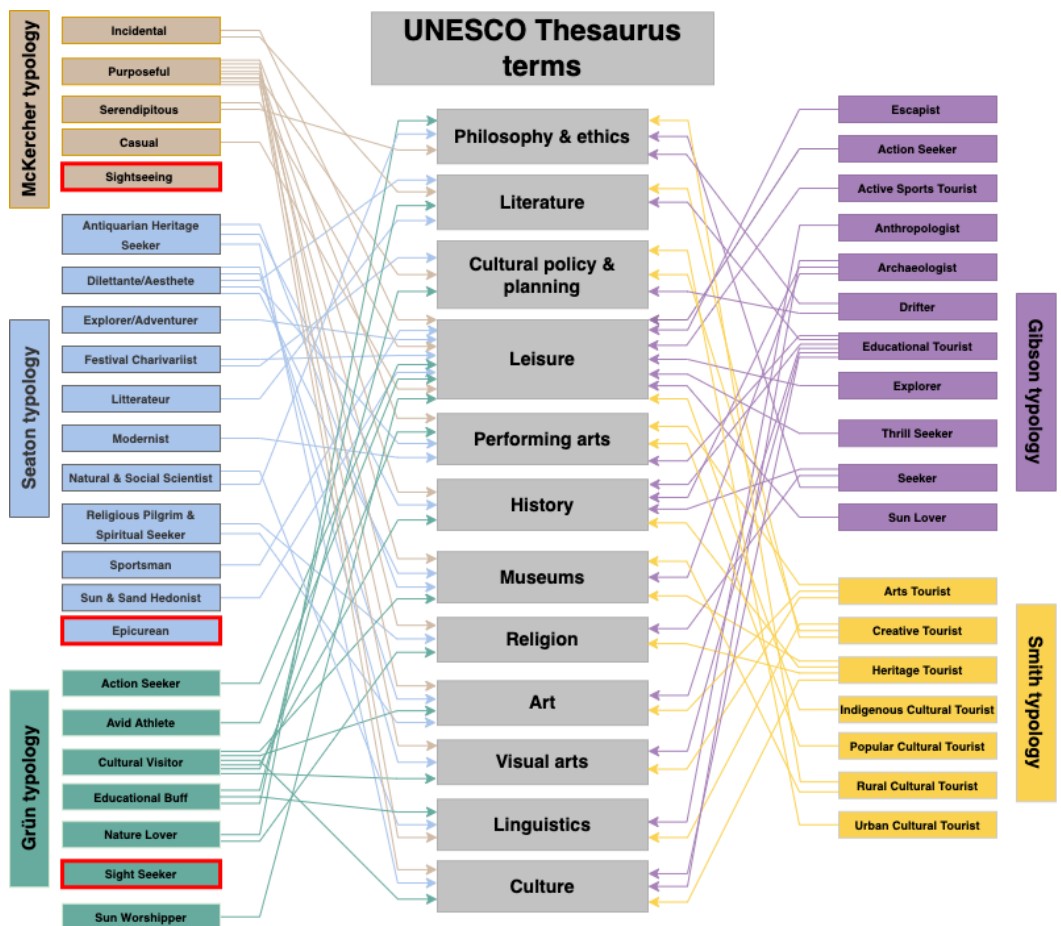

**Figure 2.** Alignment of the profiles of the five selected typologies.

| | Philosophy & ethics | Literature | Cultural policy & planning | Leisure | Performing arts | History | Museums | Religion | Art | Visual arts | Linguistics | Culture |
|---|---|---|---|---|---|---|---|---|---|---|---|---|
| **Seaton typology** | Natural & Social Scientist | Dilettante/ Aesthete Littérateur | Festival Charivariist | Explorer/ Adventurer Sun & Sand Hedonist Festival Charivariist Sportsman | Dilettante/ Aesthete Modernist | Antiquarian Heritage Seeker | Antiquarian Heritage Seeker Dilettante/ Aesthete | Religious Pilgrim & Spiritual Seeker | Dilettante/ Aesthete Religious Pilgrim & Spiritual Seeker | Dilettante/ Aesthete | Natural & Social Scientist | Antiquarian Heritage Seeker |
| **McKercher typology** | Serendipitous | Purposeful | Purposeful | Incidental Serendipitous Casual | Purposeful | Purposeful | Purposeful | Purposeful | Purposeful | Purposeful | Purposeful | Purposeful |
| **Gibson typology** | Drifter Seeker | Educational Tourist | Drifter | Escapist Action Seeker Active Sports Explorer Thrill Seeker Sun Lover | Educational Tourist | Archaeologist Educational Tourist Seeker | Archaeologist | Seeker | Educational Tourist | Educational Tourist | Anthropologist | Archaeologist Educational Tourist |
| **Smith typology** | Rural Cultural Tourist | Creative Tourist | Creative Tourist Urban Cultural Tourist | Indigenous Cultural Tourist | Arts Tourist Popular Cultural Tourist | Heritage Tourist | Heritage Tourist Rural Cultural Tourist | Heritage Tourist | Arts Tourist | Arts Tourist | Creative Tourist | Heritage Tourist |
| **Grün typology** | Nature Lover | Educational Buff | Educational Buff | Action Seeker Avid Athlete Sun Worshipper | Cultural Visitor | Cultural Visitor | Cultural Visitor | Nature Lover | Cultural Visitor | Cultural Visitor | Educational Buff | Cultural Visitor |

**Figure 3.** Overview of the aligned profiles of the five selected typologies.

As can be seen in Figure 2, in many cases, a UNESCO term was matched with multiple profiles belonging to the same typology (for example, the *History* term was matched with Gibson's *Archaeologist, Educational Tourist, and Seeker*). On the other hand, in several cases, a particular profile was matched with multiple UNESCO terms (for example, Gibson's *Seeker*

was matched with the *Philosophy and ethics, History, and Religion* terms). These observations demonstrate the complexity and multi-dimensionality of the cultural-visitor typologies and thus the need for a *harmonisation* of the existing typologies in order to satisfy the complexity of the modern cultural visitor's nature and needs.

*3.4. Clustering Stage*

In the third and final stage of the development process, which led to the specification of the ACUX profiles, a three-step clustering process was conducted:

The clustering of the profiles aligned in the *ALIGNMENT* stage in Section 3.4.1;

The meta-clustering of the clusters produced in the previous step in Section 3.4.2;

The clustering of the profiles that remained unaligned during the *ALIGNMENT* stage in Section 3.4.3.

3.4.1. Clustering of Aligned Profiles

In the first step of the *CLUSTERING* stage, twelve clusters were defined, one for each *UNESCO Thesaurus* term, and each cluster was populated with the profiles that it was matched with during the *ALIGNMENT* stage. The twelve clusters are depicted in Figure 4 (the font colouring of the profiles indicates their origin–typology).

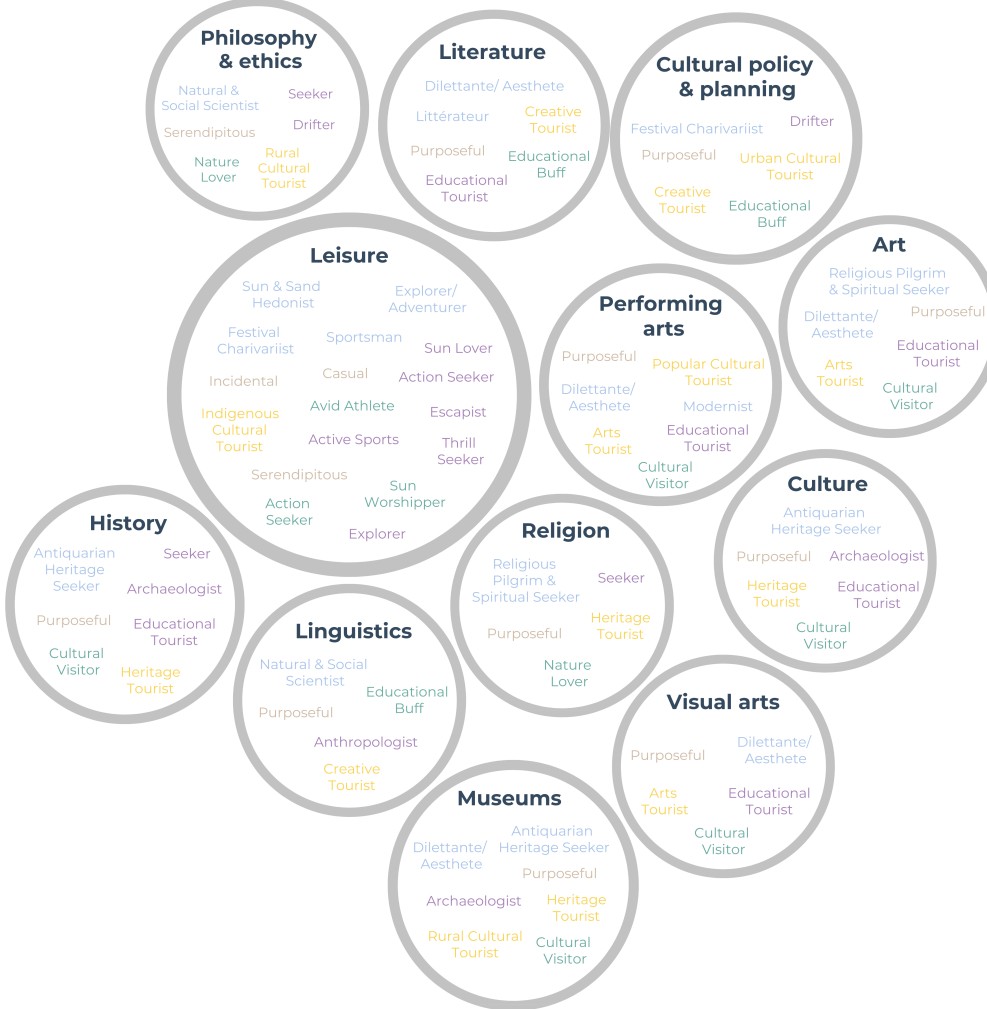

**Figure 4.** Clusters of aligned profiles.

3.4.2. Meta-Clustering of the Aligned Profile Clusters

In the second clustering step, the twelve clusters produced in the previous step were merged into six broader clusters in Figure 5, based on thematic similarity. These new broader clusters formed the first six profiles of the ACUX typology. In particular:

- The *Performing arts, Art,* and *Visual arts* clusters were merged into the ***Art Seeker*** profile.
- The *History, Museums, Literature,* and *Culture* clusters were merged into the ***Archaeologist*** profile.
- The *Religion* cluster formed the ***Religious Seeker*** profile.
- The *Leisure* cluster formed the ***Leisure Seeker*** profile.
- The *Philosophy and ethics* cluster formed the ***Naturalist*** profile.
- The *Linguistics* and *Cultural policy and planning* clusters were merged into the ***Traditionalist*** profile.

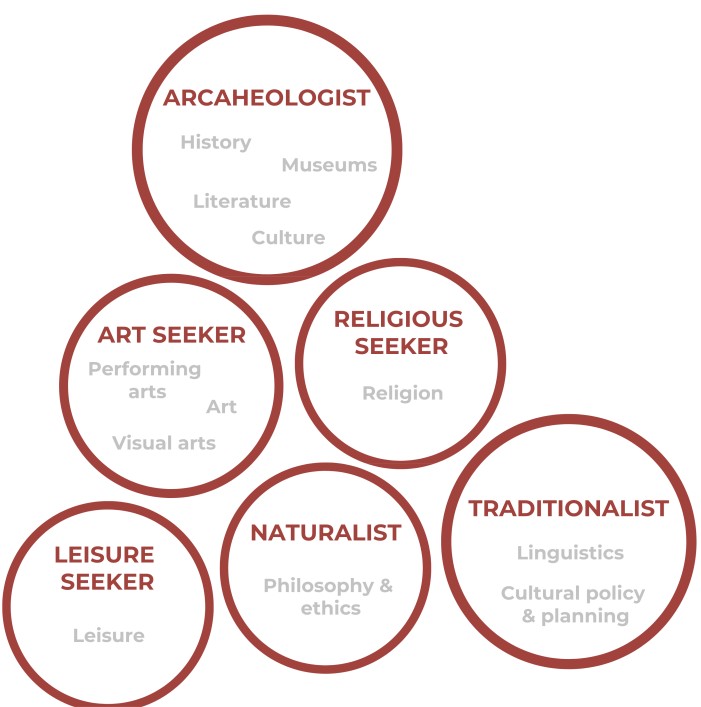

**Figure 5.** Meta-clustering of the aligned profile clusters.

3.4.3. Clustering of Unaligned Profiles

In the third and last clustering step, the three profiles that had remained unaligned during the *ALIGNMENT* stage were clustered based on the similarity of their descriptions. As a result, two more clusters were specified which in turn generated two additional ACUX profiles in Figure 6. Specifically:

- Seaton's *Epicurean* (which describes visitors committed to sensual enjoyment and particularly exquisite gastronomy) formed the ***Gourmand*** profile.
- Grün's *Sight Seeker* and McKercher's *Sightseeing* (which describe visitors who are essentially motivated by modernity) were clustered, generating the ***Viral Seeker*** profile.

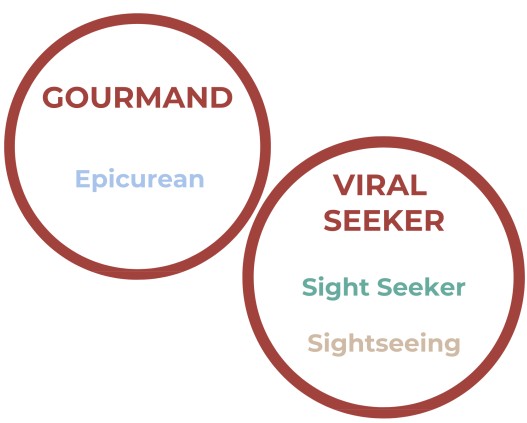

**Figure 6.** Clusters of unaligned profiles.

*3.5. The ACUX Typology*

Figure 7 illustrates the eight profiles of the proposed typology, along with the corresponding profiles of the five typologies from which the ACUX profiles were derived as a result of the *harmonisation* process.

| Seaton typology | McKercher typology | Gibson typology | Smith typology | Grün typology | ACUX |
|---|---|---|---|---|---|
| Dilettante/Aesthete Antiquarian Seeker Littérateur | Purposeful | Archaeologist Educational Tourist Seeker | Heritage Tourist Rural Cultural Tourist | Cultural Visitor | **ARCHAEOLOGIST** |
| Dilettante/Aesthete Littérateur Modernist Religious Pilgrim & Spiritual Seeker | Purposeful | Educational | Arts Tourist Popular Cultural Tourist | Cultural Visitor | **ART SEEKER** |
| Explorer/Adventurer Festival Charivariist Sportsman Sun & Sand hedonist | Incidental Serendipitous Casual | Thrill Seeker Explorer Active Sports Tourist Sun Lover Action Seeker | Indigenous Tourist | Sun Worshipper Avid Athlete Action Seeker | **LEISURE SEEKER** |
| Religious Pilgrim & Spiritual Seeker | Purposeful | Seeker | Heritage Tourist | Nature Lover | **RELIGIOUS SEEKER** |
| Natural & Social Scientist | Serendipitous | Drifter Seeker | Rural Tourist | Nature Lover | **NATURALIST** |
| Natural & Social Scientist Festival Charivariist | Purposeful | Anthropologist Drifter | Creative Tourist Urban Tourist | Educational Buff | **TRADITIONALIST** |
| Epicurean | Incidental | Action Seeker | Urban Tourist | Action Seeker | **GOURMAND** |
| Modernist | Sightseeing | Action Seeker | Urban Tourist Popular Tourist | Sight Seeker | **VIRAL SEEKER** |

**Figure 7.** The ACUX typology (along with the five typologies from which it is derived).

In the final step of the development process, the verbal specification of each ACUX profile was formulated by integrating the descriptions of the corresponding profiles of the five typologies that took part in the *harmonisation* process into a single, unified description. For example, the description of the *Naturalist* ACUX profile was synthesised from the descriptions of the (aligned) profiles *Serendipitous, Drifter, Seeker, Rural Tourist*, and *Nature Lover*. Below, the descriptions of the eight proposed profiles are presented:

1. *Archaeologist*: The *Archaeologist* visits a cultural space to gain knowledge on the ancient history and the cultural heritage of a destination, as an expert on antiquities and artefacts of the past. The *Archaeologist* is primarily interested in the classical past, history and archaeology, with a particular interest in visiting museums, galleries,

cultural sites and landscapes, archive and manuscript centres, castles, palaces, country houses, architectural sites, museums and religious sites.

2. *Art Seeker*: The *Art Seeker* feels a close tie between art and literature content, and their professional or hobbyist passions. The *Art Seeker* is the primary consumer of fine arts museums, art galleries, pottery and photography museums, theatres, concert halls, cultural centres, engraving, graffiti sites, textiles, pottery, painting, sculpture, iconography, handcrafts and literary sites.

3. *Leisure Seeker*: The primary emphasis of the *Leisure Seeker* while on a visit is staying active and fit mainly by engaging in sports but also in other recreational activities. Culture does not play a significant role when visiting a destination. Even though the *Leisure Seeker* might visit one, their participation in cultural events is minimal. Rather, they prefer to visit music festivals, sporting events, carnivals, zoos, seaside resorts, indoor game sites, health resorts, hill tribes, deserts or mountains etc.

4. *Religious Seeker*: The *Religious Seeker* is in constant search of spiritual/esoteric knowledge in order to comprehend the very meaning of life. In this context, the *Religious Seeker* visits pilgrimage destinations, cathedrals, temples etc, seeking spiritual retreats for spiritual enlightenment through religion.

5. *Naturalist*: The *Naturalist* loves to explore peaceful and spiritual places, and immerse in the natural environment, having a strong interest in rural heritage or being an eco-tourist. Villages, farms, national parks, gardens, caverns, woods, lakes, ecomuseums, cultural landscapes, and wine routes also fall within the purview of the *Naturalist*.

6. *Traditionalist*: The *Traditionalist* places a strong emphasis on custom, convention, and tradition. A typical *Traditionalist* might try to experience traditional cuisine, learn a different language in a village setting, buy arts and crafts, witness traditional cultural performances, enjoy the beautiful scenery and the laughter of joyful villagers, visit tribal communities, ethnic groups, minority cultures, and attend traditional music events.

7. *Gourmand*: Being a bon viveur, gastronome and/or wine buff, the *Gourmand* is devoted to sensual enjoyment, especially of fine food and drink. They prefer to travel first-class, stay in selected hotels, and definitely enjoy fine dining, wine tasting, food sampling, cookery courses, breweries and distilleries.

8. *Viral Seeker*: The *Viral Seeker* is motivated to travel by perceiving modernity as essential and by visiting a destination's highlights and most important landmarks. The *Viral Seeker's* satisfaction primarily derives from the mere fact of having "been there and done that". As such they are particularly interested in modern popular culture, pop concerts, shopping malls, media and film sets, technology, industrial heritage sites, fashion shows, and design museums.

## 4. Evaluation

The proposed typology has been evaluated in juxtaposition with the five typologies it derived from, using a cultural-visitor recommender and a dataset provided by the *TripMentor Project* [36]. In the context of the *TripMentor Project*, user evaluations (in the form of numerical ratings) on a plethora of cultural destinations in the Attica region of Greece (including the capital city of Athens) have been collected through the *TripAdvisor* platform. The subset of the data used in the evaluation experiment involves 9668 distinct users who have provided a total of 80,011 ratings of 5515 different places through the *TripAdvisor* platform. Therefore, the data density, defined by the ratio of the number of ratings over the number of users times the number of items is 0.15%, which designates a sparse dataset.

In cases of sparse datasets like the one described above, Factorisation Machines (FM) [37] are a family of recommendation algorithms that are known to exhibit satisfactory performancee [37–39]. Like Matrix Factorisation (MF) methods [40], FMs are generic supervised learning models that map arbitrary, real values (e.g., a user evaluation of a cultural place in the form of a numeric rating) into a low-dimensional latent feature space. In particular, FMs represent user-item interactions as a tuple of real-valued feature vectors on the one hand and a numeric target value (the rating) on the other. Those real-valued

feature vectors correspond to the latent user and item representations, and may be extended by user/item attributes and possibly other contextual features relevant to the interaction. Next, the FM model computes n-way interactions between the user and item features. The most common is the second-order model, which includes single features as well as simple pairwise feature combinations. The particular second-order model considered in the current work is based on the LightFM implementation [41].

In the experiments that follow, the user latent vector has been extended to include the five selected typologies along with the proposed typology. For example, when conducting experiments with Seaton's typology [11], each user's latent vector has been extended with 11 binary features, indicating whether the user in question has been characterised as *Antiquarian Heritage Seeker, Dilettante/aesthete*, etc. User characterisations result from the places they have evaluated, such as in the TripMentor dataset, where every point-of-interest (POI) is characterised by a number of themes. As a consequence, an extensive mapping between TripMentor themes and the typologies has been performed in the context of the current work and is available as supplementary material. Naturally, users may be assigned to more than one *UNESCO Thesaurus* term (e.g., Seaton's *Antiquarian Heritage Seeker* and *Dilettante/Aesthete* are both matched with the *Museums* term), based on the POIs they have rated.

Figures 8 and 9 outline the performance results for the five typologies considered in this work, on a set of two popular offline evaluation metrics. The first one (Figure 8) is the average precision (AP) for a list of three items. Precision is an information retrieval metric defined as the ratio of relevant recommended items to a user over the total recommended items to that user (in our case, three), averaged over all users. The second (Figure 9) is the mean reciprocal rank (MRR), again for a list of three items. The reciprocal rank is a rank accuracy metric defined as the multiplicative inverse of the rank of the first correctly recommended item, averaged over all users. All typologies have been incorporated in the LightFM algorithm [41], for a latent user/item feature space of 20 features, trained using the AdaGrad optimiser [42] for 20 epochs, with a learning rate of 0.05 and the default values of $\rho$ and $\epsilon$, using the bayesian pairwise loss function. The TripMentor dataset has been randomly split 10 times on a training (80%) and a test dataset (20%), with the results appearing on both Figures being an average of the 10 runs of the algorithm. Finally, the statistical significance of the presented results is assessed by the Student's *t*-test with $p < 0.1$.

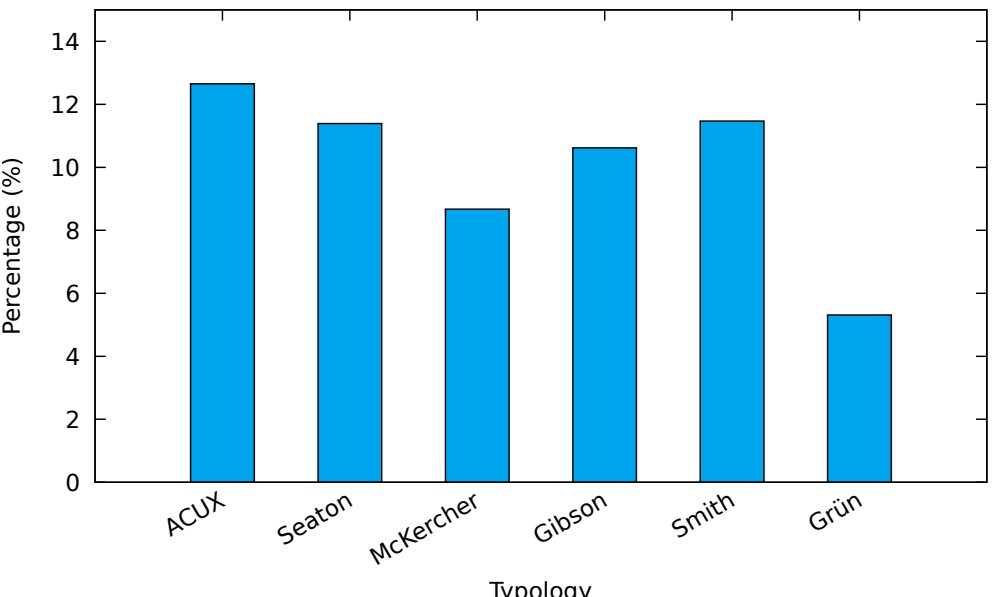

**Figure 8.** Average Precision (list of 3 recommended items).

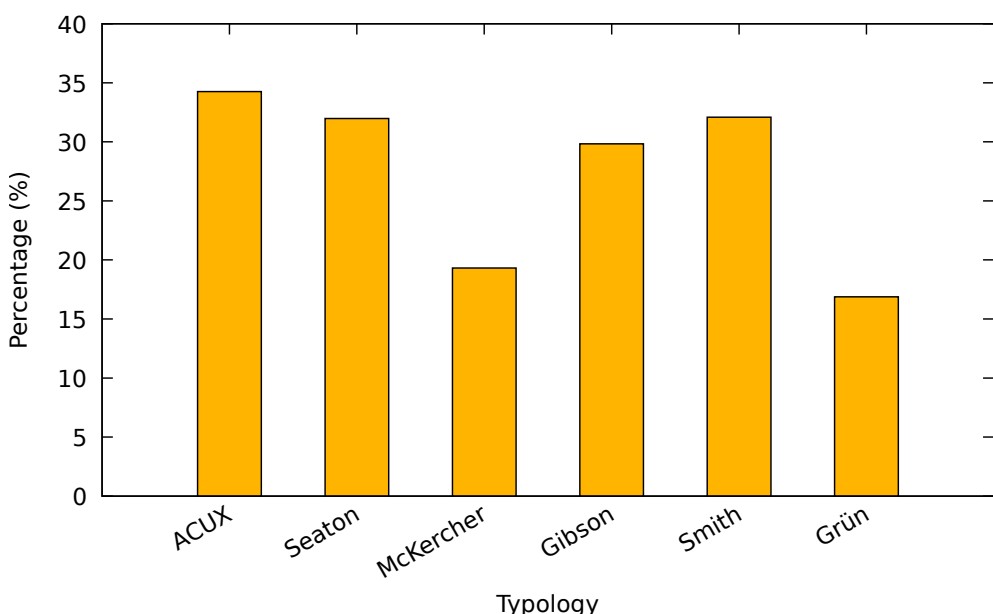

**Figure 9.** Mean Reciprocal Rank (list of 3 recommended items).

## 5. Conclusions and Future Research

In this paper, we proposed the ACUX (Augmented Cultural User eXperience) typology for classifying visitors of cultural destinations. The proposed typology aims to provide the *multi-profile* classification of cultural visitors based on their visiting preferences. Methodology-wise, the proposed typology was the output of a *harmonisation* process of selected cultural-visitor typologies that base their classification on visiting preferences. The proposed typology was evaluated in comparison with the five typologies from which it was derived through an experiment conducted using a recommender and a dataset of *TripAdvisor* user responses.

The evaluation showed that when paired with the LightFM algorithm, the proposed typology surpasses all others in terms of average precision. It clearly outperforms McKercher's and Grün's typologies by a large margin because of the cultural centrality of those typologies, and Seaton's, Gibson's, and Smith's by a smaller margin. In other words, the *harmonisation* of the existing typologies and their encapsulation into a single, unified one (ACUX) achieved more accurate profiling of cultural visitors, enabling them to reduce information overload by directly suggesting content that is more likely to meet their diverse preferences and needs.

Future research will concentrate on the development of a mobile application for cultural visitors, which will exploit the ACUX typology with the objective of providing *multi-profile* recommendations. In addition, social media tools and crowdsourcing techniques may be utilised in order to boost the visitors' prior knowledge and motivation to visit a cultural destination, further improving the usability and effectiveness of the application.

**Supplementary Materials:** The following supporting information can be downloaded at: https://www.mdpi.com/article/10.3390/digital2030020/s1, An extensive mapping between TripMentor themes and the typologies has been performed in the context of the current work.

**Author Contributions:** Conceptualisation, M.K.; methodology, M.K. and Y.C.; software, G.A.; validation, M.K., Y.C., A.T. and G.A.; formal analysis, M.K. and A.T.; investigation, M.K.; resources, M.K.; data curation, M.K., Y.C. and G.A.; writing—original draft preparation, M.K.; writing—review and editing, M.K., Y.C. and G.C.; visualisation, M.K. and A.T.; supervision, Y.C. and G.C.; project administration, M.K.; funding acquisition, M.K. All authors have read and agreed to the published version of the manuscript.

**Funding:** This research received no external funding.

**Institutional Review Board Statement:** Not applicable.

**Informed Consent Statement:** Not applicable.

**Data Availability Statement:** Not applicable.

**Conflicts of Interest:** The authors declare no conflict of interest.

## Abbreviations

The following abbreviations are used in this manuscript:

| | |
|---|---|
| ACUX | Augmented Cultural User eXperience |
| AP | Average Precision |
| CH | Cultural Heritage |
| CUX | Cultural User eXperience |
| EKT | Greek National Documentation Center |
| FM | Factorisation Machines |
| HCI | Human Computer Interaction |
| MAP | Mean Reciprocal Rank |
| MF | Matrix Factorisation |
| POI | Point-of-Interest |
| RS | Recommendation Systems |
| UI | User Interface |
| UX | User eXperience |

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
