# Peer review of "ACUX Typology: A Harmonisation of Cultural-Visitor Typologies for Multi-Profile Classification"

_digital, doi:10.3390/digital2030020_

Round 1
Reviewer 1 Report
An interesting material that studies cultural tourism from the perspective of a comparative analysis of the typologies of tourists (and their motivations), adding some quite surprising results on the table, results that can be used for further research.
Author Response
Dear Reviewer,
Thank you for the opportunity to revise our manuscript “ACUX typology: A Harmonisation of Cultural Visitor Typologies for Multi-personae Recommendation”. We appreciate the careful review and constructive suggestions. It is our belief that the manuscript is substantially improved after making the suggested edits, highlighted within the document by using coloured text (red).
Reviewer 2 Report
The paper is very interesting and well written. The introduction to the subject is adequate and the related work given is interesting. The structure of the paper is also very good. The paper could be slightly improved by:
1) Changing the last section to Discussion & Conclusions and providing a more detailed discussion of the conclusions
2) As the systems seems very interesting I would like to see more info about future work in the last paragraph of the paper, eg evaluation with real users etc.
Author Response
Dear Reviewer,
Thank you for the opportunity to revise our manuscript “ACUX typology: A Harmonisation of Cultural Visitor Typologies for Multi-personae Recommendation”. We appreciate the careful review and constructive suggestions. It is our belief that the manuscript is substantially improved after making the suggested edits, highlighted within the document by using coloured text (red).
The article should be revised as follows:
- Changing the last section to Discussion & Conclusions and providing a more detailed discussion of the conclusions
The aforementioned changes have been addressed in the revised manuscript by changing the section to "Conclusions and Future Work” and adding the proper text. The Discussion issues are presented at the Evaluation section.
- As the systems seem very interesting, I would like to see more information about future work in the last paragraph of the paper, e.g. evaluation with real users etc.
The aforementioned changes have been addressed in the revised manuscript. We remind the reviewers that during our research we conducted an evaluation experiment using a dataset of TripAdvisor user responses on a recommendation system that uses the LightFM algorithm. Therefore, the evaluation was carried out on real users indeed.
Reviewer 3 Report
It is an interesting paper. Motivation is described well. Organisation is good. Related works are fair. Technical contribution is clear.
Author Response

(The authors gave the same response as above.)

Reviewer 4 Report
The paper presents the ACUX (Augmented Cultural User eXperience) typology based on visitor preferences for multi-person recommendations, through clustering analysis
and alignment of existing typology profiles.
The topic is interesting as well as the methodology adopted.
What would require more information is paragraph 3.2: how the UNESCO categories and subcategories of visitor preferences are "translated" into "useful" categories and subcategories. What is the degree of automation? Is it done manually, or semi-automatically? And how is this "translation" done?
Similarly for ACUX clustering and alignment (3.4). How is it done? More information on the methodology should be included in the article, improving these parts.
Author Response
Dear Reviewer,
Thank you for the opportunity to revise our manuscript “ACUX typology: A Harmonisation of Cultural Visitor Typologies for Multi-personae Recommendation”. We appreciate the careful review and constructive suggestions. It is our belief that the manuscript is substantially improved after making the suggested edits, highlighted within the document by using coloured text (red).
The article should be revised as follows:
- What would require more information in paragraph 3.2: how the UNESCO categories and subcategories of visitor preferences are "translated" into "useful" categories and subcategories. What is the degree of automation? Is it done manually, or semi-automatically? And how is this "translation" done?
The aforementioned changes have been addressed in the revised manuscript. More specifically, we update section 3.2 with additional text in the first two paragraphs.
- Similarly, for ACUX clustering and alignment (3.4). How is it done? More information on the methodology should be included in the article, improving these parts.
The aforementioned changes have been addressed in the revised manuscript. More specifically, we first change section 3.4 into section 4 and then update section 4 with additional text after and before Figure 4.